# Phase Space Spin-Entropy

**DOI:** 10.3390/e26050372

**Published:** 2024-04-28

**Authors:** Davi Geiger

**Affiliations:** Courant Institute of Mathematica Sciences, New York University, New York, NY 10012, USA; dg1@nyu.edu

**Keywords:** spin entropy, phase space, quantum information, entanglement, geometric quantization

## Abstract

Quantum physics is intrinsically probabilistic, where the Born rule yields the probabilities associated with a state that deterministically evolves. The entropy of a quantum state quantifies the amount of randomness (or information loss) of such a state. The degrees of freedom of a quantum state are position and spin. We focus on the spin degree of freedom and elucidate the spin-entropy. Then, we present some of its properties and show how entanglement increases spin-entropy. A dynamic model for the time evolution of spin-entropy concludes the paper.

## 1. Introduction

This paper addresses the problem of quantifying the randomness associated with a spin-state. Our broader motivation is to study the role of randomness in quantum physics. We follow the geometric quantization (GQ) method [1,2,3,4,5] and their references, which is a formulation of quantum physics derived from classical physics in phase space. One can then view quantum physics as an information theory that incorporates randomness into the phase space of classical physics. Previous work addressed information theory aspects of quantum physics; see [6,7,8,9,10,11,12,13,14,15,16] and their references. We point out the GK-Law [15], a hypothesis stating that in a closed physical system information cannot be gained. This law also suggests a mechanism in nature that prevents a state from having its entropy decrease, which involves annihilating the state (and the particles associated with it) while still satisfying the conservation laws by creating an appropriate new state with new particles and higher entropy. Thus, the entropy of a quantum state could—and should—play a role in helping us make predictions about particle physics.

The degrees of freedom (DOFs) of a quantum state are associated with the position and spin values of a particle, i.e., all randomness of a quantum state is captured by Born’s rule in phase space once the DOFs are specified. We point out that in quantum physics, von Neumann entropy [6] quantifies only the lack of knowledge an observer has about a quantum state, i.e., it quantifies only the randomness in specifying its DOFs. Thus, for all pure states, the von Neumann entropy is zero, while our entropy formulation is focused on pure states. We will also extend our formulation to mixed states and show that the von Neumann entropy is a lower bound to our proposed entropy.

With respect to the position DOF, Geiger and Kedem [15] formulated the *x*-*p* phase space quantum entropy. Here, we focus on the spin DOF.

Note that quantifying the randomness of the spin-state along the *z*-direction alone is not sufficient, since the measurement of an eigenstate of $S_{z}$ is certain to yield the eigenvalue of this eigenstate. However, there is still randomness associated with any spin measurement in the *x*-*y* plane as demonstrated by the Stern–Gerlach experiments [17]. It is in the spin phase space where all the randomness of a spin-state is captured.

Historically, spin was first conceived as a quantum concept, but GQ derives it as the quantization of the classical two-sphere, $S^{2}$. In this Hilbert space, two independent variables, e.g., the zenith and azimuth angles, θ and ϕ, respectively, are required to specify a spin wave function ψ(θ,ϕ). These classical variables are associated with the polarization of spin particles. Not only does GQ provide a description of spin that originates from classical ideas, but it also delineates the spin phase space where the wave function and all its inherent randomness are defined.

The information, or lack thereof, associated with the phase space wave function ψ(θ,ϕ) is captured by differential entropy [18], i.e.,
(1)$$\begin{array}{c} S\left(\psi \right)=-\int_{0}^{2\pi }\int_{0}^{\pi }{\left|\psi \left(\theta ,\phi \right)\right|}^{2}\;\ln{\left|\psi \left(\theta ,\phi \right)\right|}^{2}\;\sin\theta \;d\theta \;d\phi \;. \end{array}$$We refer to (Equation 1) as spin-entropy. While many of the properties exhibited by the spin-entropy also hold for the more general Rényi entropy of the order α given by
(2)$$\begin{array}{c} S_{\alpha }\left(\psi \right)=\frac{1}{1-\alpha }\log\left({\int |\psi \left(\theta ,\phi \right)|}^{2\alpha }\;\sin\theta \;d\theta \;d\phi \right)\;, \end{array}$$
we focus on the spin-entropy in this paper.

### 1.1. Previous Work

Wehrl entropy [19] was introduced to approximate a classical entropy for a quantum state, and Lieb studied spin-coherent states to evaluate Wehrl spin-entropy [20,21,22]. Like in the case of spatial coherent states, spin coherent states constitute an overcomplete set of states. Since Wehrl entropy is based on an overcomplete basis representation, projections on this basis lead to quasi-probabilities that violate the Kolmogorov third axiom. The overcomplete basis decomposition and the arbitrariness in the choice of the spin-coherent state basis used to define the probability distribution prevent Wehrl entropy from accurately quantifying the randomness associated with spin observables.

This work starts from a similar view as [23]; here, we steer it differently, correcting their attempt to decompose the wave function in phase space.

### 1.2. Paper Organization

The paper is organized as follows: Section 2 provides a summary of GQ concepts applied to the sphere and focuses on a derivation of the wave function in spin phase space; it is based on [5]. Extensions to these GQ concepts are also in Appendix A and the references. Section 3 derives the main spin-entropy properties, including a formula for all the eigenstates of the *z*-direction for any spin, *s*. Section 4 extends the spin-entropy to mixed states and shows that von Neumann entropy is a lower bound for it. Section 5 analyzes the connection between phase space entanglement and spin-entropy. We show that the more entanglement between two particles, the larger the spin-entropy. Section 6 illustrates how a Hamiltonian model for a spin interaction causes the spin-entropy evolution to oscillate. Section 7 concludes the paper.

## 2. Some Geometric Quantization Concepts

We first provide a summary of concepts used in this paper developed by the GQ of the two-sphere $S^{2}$ to describe spin operators and eigenfunctions in phase space. Some extra material to complete the descriptions presented in this summary is presented in the Appendix A.

### 2.1. Complex Plane and the Sphere

Consider the local variable z=x+iy to describe the complex plane ${\mathrm{CP}}^{1}$. The stereographic projection of the 3D embedding of the sphere $S^{2}$ of radius *s* via the south pole Z=−1, to the plane Z=0, is given as follows:(3)$$\begin{array}{c} X=s\;\sin\theta \cos\phi =s\frac{z+z^{*}}{1+zz^{*}},\;Y=s\;\sin\theta \sin\phi =-is\frac{z-z^{*}}{1+zz^{*}},\;Z=s\;\cos\theta =s\frac{1-zz^{*}}{1+zz^{*}} \end{array}$$
where (θ,ϕ)∈[0,π)×[0,2π) are the zenith and azimuth spherical angles, respectively. The inverse mapping is then given by the following:(4)$$\begin{array}{c} z=\frac{X+iY}{1+Z}=\tan\frac{\theta }{2}\;e^{i\phi }\;\to z^{*}=\frac{X-iY}{1+Z}=\tan\frac{\theta }{2}\;e^{-i\phi }\;. \end{array}$$
and the mapping of the south pole Z=−1 to *z* requires the stereographic projection via the north pole, i.e., $z=\frac{X+iY}{1-Z}$.

### 2.2. Symplectic Structure for the Sphere and Canonical Transformations

A symplectic structure for the sphere manifold is given by the following:(5)$$\begin{array}{c} \Omega =2\;i\;s\;\frac{dz\wedge dz^{*}}{{\left(1+zz^{*}\right)}^{2}}=s\sin\theta \;d\theta \wedge d\phi \end{array}$$
where 2s is an integer. Canonical transformations preserve Ω. Consequently, for vector fields $\xi =(\xi^{z},\xi^{z^{*}})$ to be generators of canonical transformations in the sphere, the transformation $\vec{z}=\left(z,z^{*}\right)\to \vec{z}+\xi \left(\vec{z}\right)$ must satisfy the following:(6)$$\begin{array}{cc} 0=\delta_{\xi }\Omega & =\frac{1}{2}\Omega_{\mu ,\nu }\left(\vec{z}+\xi \right)d\left(z^{\mu }+\xi^{\mu }\left(\vec{z}\right)\right)\;\wedge \;d\left(z^{\nu }+\xi^{\nu }\left(\vec{z}\right)\right)-\frac{1}{2}\Omega_{\mu ,\nu }\;dz^{\mu }\;\wedge \;dz^{\nu }\\ \; & =\frac{1}{2}\left[\partial_{z^{\mu }}\left(\xi^{\alpha }\Omega_{\alpha ,\nu }\right)-\partial_{z^{\nu }}\left(\xi^{\alpha }\Omega_{\alpha ,\mu }\right)\right]\;dz^{\mu }\;\wedge \;dz^{\nu }\\ \; & =\partial (\xi^{\alpha }\Omega_{\alpha ,\nu }\;dz^{\nu })\;. \end{array}$$Thus, for every infinitesimal canonical transformation, we can associate a function *J* on $S^{2}$, such that $-dJ=\xi^{\alpha }\Omega_{\alpha ,\nu }\;dz^{\nu }$ is a closed one-form. Also, by inverting Ω, we obtain the following:(7)$$\begin{array}{c} \xi^{\mu }\Omega_{\mu ,\nu }=-\partial_{z^{\nu }}J\;\to \;\xi^{\mu }=\Omega^{\mu ,\nu }\partial_{z^{\nu }}J\;. \end{array}$$Thus, conversely, given a classical observable, *J*, we can readily obtain the generators ξ of the infinitesimal canonical transformation.

### 2.3. Spin Operator and Eigenfunctions

In order to derive the spin operators $S_{x},S_{y},S_{z}$, one must consider the classical functions $J_{X}=X\left(z,z^{*}\right)$, $J_{Y}=Y\left(z,z^{*}\right)$, $J_{Z}=Z\left(z,z^{*}\right)$ in the sphere, as described in (Equation 3). The generators of the canonical transformations associated with these functions, derived from (Equation 7), are isometries of the sphere. In Appendix A, we present a summary of the GQ steps to obtain the quantum operator operators (Equation 56) acting on the two-sphere $S^{2}$ associated with these functions $J_{X},J_{Y},J_{Z}$. These are the spin operators and can then be written as follows:(8)$$\begin{array}{cc} S_{X}\; & =\frac{\left(1-z^{2}\right)}{2}\partial_{z}+s\frac{1}{\left(1+zz^{*}\right)}\left(z+\frac{\left(z^{*}+z^{2}z^{*}\right)}{2}\right)\\ S_{Y} & =i\frac{\left(1+z^{2}\right)}{2}\partial_{z}-is\;\frac{1}{\left(1+zz^{*}\right)}\left(z-\frac{\left(z^{*}-z^{2}z^{*}\right)}{2}\right)\\ S_{Z} & =-z\partial_{z}+s\;\frac{1}{\left(1+zz^{*}\right)} \end{array}$$The eigenfunctions of the operator $S_{Z}$ (due to the polarization condition, see Appendix A (Equation 54), (Equation 55)), are of the following form:(9)$$\begin{array}{c} \langle z,z^{*}|\xi_{s,m}\rangle =\Psi_{s,m}\left(z,z^{*}\right)=\frac{1}{{\left(1+zz^{*}\right)}^{s}}f_{s,m}\left(z\right) \end{array}$$
where $|\xi_{s,m}\rangle$ is the spin eigenstate along the *Z*-axis. We can then derive the holomorphic function $f_{s,m}\left(z\right)$ as follows:(10)$$\begin{array}{cc} S_{Z}\Psi_{s,m}\left(z,z^{*}\right) & =sz\frac{z^{*}}{{\left(1+zz^{*}\right)}^{1+s}}f_{s,m}\left(z\right)-\frac{z}{{\left(1+zz^{*}\right)}^{s}}\partial_{z}f_{s,m}\left(z\right)+s\frac{1}{{\left(1+zz^{*}\right)}^{1+s}}f_{s,m}\left(z\right)\\ \; & =\frac{1}{{\left(1+zz^{*}\right)}^{s}}\left(-z\partial_{z}+s\right)f_{s,m}\left(z\right)\\ \; & =m\Psi_{s,m}\left(z,z^{*}\right)\\ \; & ⇓\;\mathrm{applying}\;(9)\\ \left(-z\partial_{z}+s\right)f_{s,m}\left(z\right) & =mf_{s,m}\left(z\right) \end{array}$$
with solutions of the following form:(11)$$\begin{array}{cc} f_{s,m}\left(z\right) & =\frac{1}{\sqrt{4\pi }}\sqrt{\left(2s+1\right)\;C_{s+m}^{2s}}\;\;z^{s-m}\;, \end{array}$$
where $C_{p}^{n}=\frac{n!}{p!\;(n-p)!}$, and we note that, compared to the presentation in Nair [5], the functions $f_{s,m}\left(z\right)$ have an extra factor $\frac{1}{\sqrt{4\pi }}$ for normalization purposes. Using the representation (Equation 4), the wave function (Equation 9) is written as follows:(12)$$\begin{array}{cc} \Psi_{s,m}\left(\theta ,\phi \right)\; & =\frac{\sqrt{\left(2s+1\right)}}{2^{s}\;\sqrt{4\pi }}\;{\left[C_{s+m}^{2s}\right]}^{\frac{1}{2}}{\left(1+\cos\theta \right)}^{\frac{s+m}{2}}{\left(1-\cos\theta \right)}^{\frac{s-m}{2}}\;e^{i(s-m)\;\phi }\;. \end{array}$$These wave functions form an orthonormal basis, i.e.,
(13)$$\begin{array}{cc} \int_{S^{2}}\;\frac{\Omega }{s}\;\Psi_{s,m}\;\Psi_{s,m^{\prime }}^{*} & =i\int_{S^{2}}\;\frac{dz\wedge dz^{*}}{2\pi {\left(1+zz^{*}\right)}^{2s+2}}\;f_{s,m}^{*}\left(z\right)f_{s,m^{\prime }}\left(z\right)\\ \; & =\frac{1}{2^{2s+1}}\int_{0}^{2\pi }\frac{d\phi }{2\pi }\;\int_{0}^{\pi }\;\sin\theta \;d\theta \;{\left(1+\cos\theta \right)}^{2s}\;f_{s,m}^{*}\left(z\left(\theta ,\phi \right)\right)f_{s,m^{\prime }}\left(z\left(\theta ,\phi \right)\right)\\ \; & =\delta_{m,m^{\prime }}\frac{\left(2s+1\right)\;C_{s+m}^{2s}}{2^{2s+1}}\int_{0}^{\pi }\;{\left(1+\cos\theta \right)}^{s+m}\;{\left(1-\cos\theta \right)}^{s-m}\;\sin\theta \;d\theta \\ \; & =\delta_{m,m^{\prime }}\;. \end{array}$$Then, through the linear superposition of these eigenstates, all spin wave functions in phase space are constructed.

Note that a wave function in the *x*-*p* phase space is decomposed as a product of a wave function on *x* and a wave function on *p*, related to each other by the Fourier transform. In contrast, a wave function in the spin phase space cannot be reduced, i.e., cannot be decomposed as a product of wave functions where each contains just one variable of phase space. Instead, a wave function over the entire spin phase space, the entire sphere, is required.

## 3. Spin-Entropy in Phase Space

**Lemma** **1**(Space Homogeneity)**.**
*Spin-entropy, given by (Equation 1), is invariant under rotations and reflections of the coordinate system.*

**Proof.** Given a function $g:S^{2}\to R$, we define a functional on the sphere,
$$F\left(g\right)=\int_{S^{2}}g\left(z,z^{*}\right)\;d\sigma \left(z,z^{*}\right)\;,$$
where $d\sigma \left(z,z^{*}\right)=\frac{1}{s}\Omega =2i\;\frac{dz\wedge dz^{*}}{{\left(1+zz^{*}\right)}^{2}}$.The spin-entropy is one such functional F(g) for the following:
(14)$$\begin{array}{c} g\left(z,z^{*}\right)=-|\psi \left(z,z^{*}\right){|}^{2}\log{\left|\psi \left(z,z^{*}\right)\right|}^{2}\;. \end{array}$$Consider an isometry transformation $\xi :S^{2}\to S^{2}$, so $\xi \left(z,z^{*}\right)=\left(\xi^{z}\left(z,z^{*}\right),\xi^{z^{*}}\left(z,z^{*}\right)\right)$. Applying an isometry to the wave function $\psi (z,z^{*})$, i.e., applying it to *g*, leads to the following:
(15)$$\begin{array}{cc} F(g\circ \xi ) & =\int_{S^{2}}g\left(\xi (z,z^{*})\right)\;d\sigma \left(z,z^{*}\right)\\ \; & \;\;\mathrm{changing}\mathrm{coordinates}\;d\sigma \left(z,z^{*}\right)=\det J\left(\xi \right)\;d\sigma \left(\xi (z,z^{*})\right)\\ \; & =\int_{S^{2}}g\left(\xi^{z}\left(z,z^{*}\right),\xi^{z^{*}}\left(z,z^{*}\right)\right)\det J\left(\xi \right)\;d\sigma \left(\xi^{z}\left(z,z^{*}\right),\xi^{z^{*}}\left(z,z^{*}\right)\right)\\ \; & \;\;\xi \;\mathrm{is}a\mathrm{canonical}\mathrm{transformation}\\ \; & =\int_{S^{2}}g\left(\xi^{z},\xi^{z^{*}}\right)\;d\sigma \left(\xi^{z},\xi^{z^{*}}\right)=F\left(g\right) \end{array}$$
where we use the fact that isometries of the sphere are canonical transformations that preserve the symplectic form, meaning the determinant of the Jacobian of canonical transformations is 1, i.e., detJ(ξ)=1.Thus, the spin-entropy is invariant under isometries. The isometries of the sphere are the three reflections, and combinations of two reflections give the rotations. □

One immediate conclusion is the homogeneity of the space due to the invariance of the spin-entropy to rotations and reflections of the coordinate system. So, given a spin-state associated with s=1, such that for some choice of the *z*-coordinate, it has m=1, this state will have a spin-entropy value, regardless of whether one chooses this *z*-coordinate to describe it.

Let us consider the eigenstates along the *z*-direction, described by (Equation 12). The spin-entropy (Equation 1) associated with these eigenfunctions is as follows:(16)$$\begin{array}{cc} S(\psi_{s,m}) & =-\int_{0}^{2\pi }\;\int_{0}^{\pi }\left(\frac{{\left(1+\cos\theta \right)}^{s+m}{\left(1-\cos\theta \right)}^{s-m}}{2\pi \;Z(s,m)}\right)\;\ln\left(\frac{{\left(1+\cos\theta \right)}^{s+m}{\left(1-\cos\theta \right)}^{s-m}}{2\pi \;Z(s,m)}\right)\;\sin\theta \;d\theta \;d\phi \\ \; & =\ln\left(2\pi \;Z(s,m)\right)-\frac{1}{Z(s,m)}\int_{-1}^{1}{\left(1+u\right)}^{s+m}{\left(1-u\right)}^{s-m}\;\ln\left[{\left(1+u\right)}^{s+m}{\left(1-u\right)}^{s-m}\right]\;du\;, \end{array}$$
where the probability normalization constant is given by the following: $2\pi \;Z\left(s,m\right)=2\pi \;\frac{2^{2s+1}}{\left(2s+1\right)\;C_{s+m}^{2s}}$ and u=cosθ.

For the rest of this section, and manipulation purposes, let us define the integer variables p=s+m and q=s−m, and since −s≤m≤s, we have p,q∈[0,1,…,2s−1,2s]. We may refer to p,q,s,m as it is more convenient during manipulations.

**Lemma** **2**(Spin-entropy formula for *z*-eigenstates)**.**
*The spin-entropy in phase space for the z-eigenstates $\psi_{s,m}$ is invariant under the reflection m→−m and can be written as follows:*
(17)$$\begin{array}{c} S\left(\psi_{s,m}\right)=\ln\left(4\pi \right)-\ln C_{s+m}^{2s}+\sum_{j=1}^{s-m+1}\frac{s+m}{\left(s+m+j\right)}+\sum_{j=1}^{s+m+1}\frac{s-m}{\left(s-m+j\right)} \end{array}$$*where θ(.) is the Heaviside function, i.e., for x≥0,θ(x)=1 and for x<0,θ(x)=0.*

**Proof.** Following on (Equation 16), let us first proceed to evaluate the spin-entropy. We will refer to $S_{p,q}$ as the spin-entropy of the eigenstate $S(\psi_{s,m})$, and $Z_{p,q}=\left[\frac{2^{s+1}}{2s+1}\;\frac{p!\;q!}{2s!}\right]$ refers to the normalization Z(s,m). We then write (Equation 16) as follows:
(18)$$\begin{array}{cc} S_{p,q}\; & =\ln\left(2\pi Z_{p,q}\right)-\frac{1}{Z_{p,q}}\int_{-1}^{1}{\left(1+u\right)}^{p}\;{\left(1-u\right)}^{q}\;\ln\left({\left(1+u\right)}^{p}\;{\left(1-u\right)}^{q}\right)\;du\\ \; & =\ln\left(2\pi Z_{p,q}\right)-\frac{1}{Z_{p,q}}\left[pI_{p,q}+qI_{q,p}\right] \end{array}$$
where
(19)$$\begin{array}{cc} I_{p,q} & =\int_{-1}^{1}{\left(1+u\right)}^{p}\;{\left(1-u\right)}^{q}\;\ln\left(1+u\right)\;du\\ \; & ⇓\\ I_{q,p} & =\int_{-1}^{1}{\left(1+u\right)}^{q}\;{\left(1-u\right)}^{p}\;\ln\left(1+u\right)\;du=\int_{-1}^{1}{\left(1-u^{\prime }\right)}^{q}\;{\left(1+u^{\prime }\right)}^{p}\;\ln\left(1-u^{\prime }\right)\;du^{\prime }\;. \end{array}$$
and in the last step, we apply the transformation $u\to u^{\prime }=-u$.Noting that $Z_{p,q}=\left[\frac{2^{s+1}}{2s+1}\;\frac{p!\;q!}{2s!}\right]=Z_{q,p}$, it is then clear that the spin-entropy (Equation 18) has the property $S_{p,q}=S_{q,p}$. And since the transformation m→−m is equivalent to p↔q, we obtain $S\left(\psi_{s,-m}\right)=S\left(\psi_{s,m}\right)$, concluding the first statement of the lemma.In order to evaluate $I_{p,q}$, we define the following:
(20)$$\begin{array}{c} \left\{\begin{array}{c} U={\left(1-u\right)}^{q}\\ V=\frac{{\left(1+u\right)}^{p+1}}{\left(p+1\right)}\left[\ln\left(1+u\right)-\frac{1}{\left(p+1\right)}\right] \end{array}\right.\;\Rightarrow \;\left\{\begin{array}{c} dU=q{\left(1-u\right)}^{q-1}\;du\\ dV={\left(1+u\right)}^{p}\ln\left(1+u\right)\;du \end{array}\right. \end{array}$$Thus,
(21)$$\begin{array}{cc} I_{p,q} & =\int_{-1}^{1}{\left(1+u\right)}^{p}\;{\left(1-u\right)}^{q}\;\ln\left(1+u\right)\;du\\ \; & =\frac{{\left(1-u\right)}^{q}\;{\left(1+u\right)}^{p+1}}{p+1}\left[\ln\left(1+u\right)-\frac{1}{p+1}\right]\;{|}_{-1}^{1}-\frac{q}{p+1}\int_{-1}^{1}{\left(1-u\right)}^{q-1}{\left(1+u\right)}^{p+1}\left[\ln\left(1+u\right)-\frac{1}{p+1}\right]\;du\\ \; & =\left\{\begin{array}{cc} \frac{{\left(1+u\right)}^{2s+1}}{\left(2s+1\right)}\left[\ln\left(1+u\right)-\frac{1}{\left(2s+1\right)}\right]\;{|}_{-1}^{1}=Z_{2s,0}\left[\ln2-\frac{1}{\left(2s+1\right)}\right] & \;p=2s\;,\;q=0\\ -\frac{q}{p+1}\left(I_{p+1,q-1}-\frac{1}{p+1}Z_{p+1,q-1}\right) & \;0\leq p<2s\;,\;0<q\leq 2s \end{array}\right.\; \end{array}$$Consider an ansatz, where $I_{p,q}=Z_{p,q}F_{p,q}$. Then, the case p=2s,q=0 is already solved in (Equation 21), i.e., $I_{2s,0}=Z_{2s,0}F_{2s,0}=Z_{2s,0}\left[\ln2-\frac{1}{\left(2s+1\right)}\right]$. We can then advance (Equation 21) for 0≤p<2s,0<q≤2s by a recursive method, as follows:
(22)$$\begin{array}{cc} I_{p,q} & =Z_{p,q}F_{p,q}=-\frac{q}{\left(p+1\right)}Z_{p+1,q-1}\left(F_{p+1,q-1}-\frac{1}{\left(p+1\right)}\right)\\ \; & ⇓\;\mathrm{solving}\;\mathrm{for}\;F_{p,q}\\ F_{p,q} & =-\frac{q}{p+1}\frac{Z_{p+1,q-1}}{Z_{p,q}}\left(F_{p+1,q-1}-\frac{1}{p+1}\right)\\ \; & =F_{p+1,q-1}-\frac{1}{\left(p+1\right)}=F_{p+2,q-2}-\frac{1}{p+2}-\frac{1}{p+1}=\ldots \\ \; & =F_{p+q,0}-\sum_{j=1}^{n}\frac{1}{\left(p+j\right)},\;0\leq p<2s\;,\;\;0<q\leq 2s\\ \; & \;\mathrm{inserting}\;\mathrm{the}\;\mathrm{case}F_{p+q,0}=F_{2s,0}=\left[\ln2-\frac{1}{\left(2s+1\right)}\right]\\ \; & =\left[\ln2-\frac{1}{\left(2s+1\right)}\right]-\sum_{j=1}^{n}\frac{1}{\left(p+j\right)},\;0\leq p,q\leq 2s\\ \; & =\ln2-\sum_{j=1}^{n+1}\frac{1}{\left(p+j\right)}\;,\;0\leq p,q\leq 2s \end{array}$$We can then apply (Equation 22) to (Equation 18), while using $Z_{p,q}=Z_{q,p}$, to obtain the following:
(23)$$\begin{array}{cc} S_{p,q} & =\ln\left(2\pi \frac{q!p!}{(2s)!}\;\frac{2^{s+1}}{2s+1}\right)-\left[pF_{p,q}+qF_{q,p}\right]\\ \; & =\ln\left(4\pi \right)-\ln\left(\frac{(2s+1)!}{q!p!}\right)+\sum_{j=1}^{q+1}\frac{p}{\left(p+j\right)}+\sum_{j=1}^{p+1}\frac{q}{\left(q+j\right)}\;. \end{array}$$Replacing p=s+m and q=s−m into (Equation 23) concludes the lemma. □

The entropies for the special cases $s=0,\frac{1}{2},1$ and m=−s,−s+1,…,s are readily derived as follows:(24)$$\begin{array}{cc} S_{s=0,m=0} & =\ln(4\pi )\approx 2.5310\;.\\ S_{s=\frac{1}{2},m=\pm \frac{1}{2}} & =\frac{1}{2}+\ln\left(2\pi \right)\approx 2.3379\;.\\ S_{s=1,m=\pm 1} & =\frac{2}{3}+\ln\left(\frac{4}{3}\pi \right)\approx 2.0991\;.\\ S_{s=1,m=0} & =\frac{5}{3}+\ln\left(\frac{2}{3}\pi \right)\approx 2.4059\;. \end{array}$$

**Lemma** **3**(Spin-Entropy Variation Across *z*-Eigenstates)**.**
*Let $\psi_{s,m}$ be the z-eigenstates of spin, s. Let $m^{+}$ represent m restricted to the range $2m+1\in Z^{+}$, i.e., $\mathrm{mod}\left(s,1\right)\leq m^{+}\leq s$, where mod(s,n) is the remainder after dividing s by n. The spin-entropy $S(\psi_{s,m^{+}})$ decreases as $m^{+}$ increases. Moreover, the decrease in $S(\psi_{s,m^{+}})$, varying from $m^{+}$ to $1+m^{+}$, is given by the following:*
$$\begin{array}{cc} S\left(\psi_{s,1+m^{+}}\right)-S\left(\psi_{s,m^{+}}\right) & =\ln\left(\frac{s+1+m^{+}}{s-m^{+}}\right)-\sum_{j=s-m^{+}}^{s+m^{+}}\frac{1}{j}\;,\;m^{+}<s\\ \; & =\int_{s-m^{+}}^{s+1+m^{+}}\frac{1}{x}\;dx\;-\;\sum_{j=s-m^{+}}^{s+m^{+}}\frac{1}{j}\\ \; & <0 \end{array}$$

**Proof.** Adopting the notation p=s+m,q=s−m, we can rewrite the last two terms in (Equation 17) as follows:
(25)$$\begin{array}{c} \sum_{j=1}^{q+1}\frac{p}{\left(p+j\right)}+\sum_{j=1}^{p+1}\frac{q}{\left(q+j\right)}=\sum_{j=p+1}^{2s+1}\frac{2s}{j}+\;\theta \left(p-q-1\right)\sum_{j=q+1}^{p}\frac{q}{j} \end{array}$$
where θ(x) is the Heaviside function, i.e., θ(x)=1 for x≥0 and θ(x)=0, otherwise. Consider the difference of consecutive values of the spin-entropy described by (Equation 17) for $m\in m^{+}$, i.e.,
(26)$$\begin{array}{cc} S\left(\psi_{s,m+1}\right)-S\left(\psi_{s,m}\right) & =\ln\left(\frac{(p+1)!(q-1)!}{p!q!}\right)+\sum_{j=p+2}^{2s+1}\frac{2s}{j}-\sum_{j=p+1}^{2s+1}\frac{2s}{j}+\sum_{j=q}^{p+1}\frac{q-1}{j}-\sum_{j=q+1}^{p}\frac{q}{j}\\ \; & =\ln\left(\frac{p+1}{q}\right)-\frac{2s}{p+1}+1+\frac{q}{p+1}-\sum_{j=q}^{p+1}\frac{1}{j}\\ \; & =\ln\left(\frac{p+1}{q}\right)+\frac{1}{p+1}-\sum_{j=q}^{p+1}\frac{1}{j}\\ \; & =\ln\left(\frac{p+1}{q}\right)-\sum_{j=q}^{p}\frac{1}{j}\\ \; & =\int_{q}^{p+1}\frac{1}{x}\;dx-\sum_{j=q}^{p}\frac{1}{j}\\ \; & <0\;\mathrm{for}\;\;p\geq q\;\;\Leftrightarrow \;\;m^{+}\;\mathrm{set} \end{array}$$For the last step we used Lemma 8. □

Figure 1 illustrates Lemma 3 for s=50.

**Lemma** **4**(Negative Spin-Entropy across *z*-Eigenstates)**.**
*Let $\psi_{s,m}$ be the z-eigenstates of the spin, s. For the spin-entropy of $\psi_{s,m}$ to have a negative value, it is required that $s\geq 16\frac{1}{2}$.*

**Proof.** From Lemma 3, the lowest entropy of a spin, *s*, particle is attained for m=±s. Examining the spin-entropy formula (Equation 17) for such cases and requiring it to be negative, we obtain the following:
(27)$$\begin{array}{c} S\left(\psi_{s,m=\pm s}\right)=\ln\left(4\pi \right)-\ln\left(2s+1\right)+\frac{2s}{2s+1}<0\;\Leftrightarrow \;\ln\frac{2s+1}{4\pi }>\frac{2s}{2s+1} \end{array}$$
and since *s* must be a multiple of $\frac{1}{2}$, the minimum value of *s* for which this is possible is $s_{\min}=16\frac{1}{2}$. □

Let us examine some properties of spin $s=\frac{1}{2}$ and s=1 for all spin states.

### 3.1. Spin One-Half

**Lemma** **5**(Spin $\frac{1}{2}$ entropy)**.**
*All $s=\frac{1}{2}$ spin states have the same spin-entropy.*

**Proof.** Let us refer to |ψ〉 as an arbitrary $s=\frac{1}{2}$ spin-state. All $s=\frac{1}{2}$ spin states are reached by the application of an element of the SU(2) to |ψ〉.There exists a two-to-one homomorphic mapping of the group SU(2) onto the group SO(3). If A∈SU(2) maps onto R(A)∈SO(3), then R(A)=R(−A). This implies that the representations of SO(3) are also representations of SU(2).By Lemma 1, we conclude that all $s=\frac{1}{2}$ spin-states will be reached by a canonical transformation of any given |ψ〉 $s=\frac{1}{2}$ spin-state and, therefore, have the same spin-entropy. □

### 3.2. Spin One

The most general normalized s=1 spin-state can be written as follows:(28)$$\begin{array}{c} \Psi^{s=1}\left(\theta ,\phi \right)=e^{i\phi_{1}}\left(\cos\theta_{0}\;e^{i\phi_{0}}\;\psi_{1,0}+\sin\theta_{0}\cos\theta_{1}\psi_{1,1}+\sin\theta_{0}\sin\theta_{1}\;e^{i\phi_{-1}}\psi_{1,-1}\right) \end{array}$$
where $\psi_{1,-1},\psi_{1,0},\psi_{1,1}$ are derived from the following: (Equation 12) as follows: (29)$$\begin{array}{c} \psi_{1,1}\left(\theta ,\phi \right)=\sqrt{\frac{3}{16\pi }}\left(1+\cos\theta \right),\;\psi_{1,0}\left(\theta ,\phi \right)=\sqrt{\frac{3}{4\pi }}\;\sin\theta \;e^{i\phi },\;\psi_{1,-1}\left(\theta ,\phi \right)=\sqrt{\frac{3}{16\pi }}\left(1-\cos\theta \right)\;e^{i2\phi } \end{array}$$

**Lemma** **6.**
*The maximum value of the spin-entropy for s=1 is $S_{\max}^{1}=ln\frac{2\pi }{3}+\frac{5}{3}\approx 2.4059$, and it is reached by the state $\psi_{1,0}\left(\theta ,\phi \right)$ and all its canonical transformations (rotations and reflections). The minimum value of the spin-entropy for s=1 is $S_{\min}^{1}=\ln\frac{4\pi }{3}+\frac{2}{3}\approx 2.0991$ and it is reached by the states $\psi_{1,-1}\left(\theta ,\phi \right)$ and $\psi_{1,1}\left(\theta ,\phi \right)$ and all their canonical transformations (rotations and reflections).*


**Proof.** The probability density does not depend on the global phase term $e^{i\phi_{1}}$ and, thus, the spin-entropy varies according to the four parameters $\phi_{-1},\phi_{0},\theta_{0},\theta_{1}$. Note that for a given set of these parameters, all canonical transformations (rotations and reflections) of this wave function will correspond to transformations in parameter space.For example, for the canonical transformation of a rotation around the z-axis, we have ϕ→ϕ+δ and the wave function (Equation 28) is then written as follows:
(30)$$\begin{array}{cc} \Psi^{s=1}\left(\theta ,\phi +\delta ;\;\phi_{1},\phi_{-1},\phi_{0},\theta_{0},\theta_{1}\right) & =\sqrt{\frac{3}{16\pi }}\;e^{i\phi_{1}}\left(2\cos\theta_{0}\;e^{i\phi_{0}}\;\sin\theta \;e^{i\delta }e^{i\phi }+\sin\theta_{0}\cos\theta_{1}\left(1+\cos\theta \right)\right.\\ \; & \;\left.+\sin\theta_{0}\sin\theta_{1}\;e^{i\phi_{-1}}\left(1-\cos\theta \right)\;e^{i2\delta }e^{i2\phi }\right)\\ \; & =\sqrt{\frac{3}{16\pi }}\;e^{i\phi_{1}}\left(2\cos\theta_{0}\;e^{i(\phi_{0}+\delta )}\;\sin\theta \;e^{i\phi }+\sin\theta_{0}\cos\theta_{1}\left(1+\cos\theta \right)\right.\\ \; & \;\left.+\sin\theta_{0}\sin\theta_{1}\;e^{i(\phi_{-1}+2\delta )}\left(1-\cos\theta \right)\;e^{i2\phi }\right)\\ \; & =\Psi^{s=1}\left(\theta ,\phi ;\;\phi_{1},\phi_{-1}+2\delta ,\phi_{0}+\delta ,\theta_{0},\theta_{1}\right) \end{array}$$
corresponding to a translation in the parameters $\left(\phi_{0},\phi_{-1}\right)\to \left(\phi_{0}+\delta ,\phi_{-1}+2\delta \right)$.The rotations have three degrees of freedom and, thus, the space of parameters that cause the spin-entropy to vary is reduced to one. Thus, we examine the wave function restricted to varying one parameter.Consider the spin-1 state (Equation 28) for the cases of $\phi_{-1}=\phi_{0}=0$ and $\theta_{0}=\frac{\pi }{2}$, i.e.,
(31)$$\begin{array}{c} \Psi_{\theta_{1}}^{s=1}\left(\theta ,\phi \right)=\sqrt{\frac{3}{16\pi }}\;e^{i\phi_{1}}\left(\cos\theta_{1}\left(1+\cos\theta \right)+\sin\theta_{1}\;\left(1-\cos\theta \right)\;e^{i2\phi }\right)\;, \end{array}$$
where $\theta_{1}$ is the free parameter. All canonical transformations of such a state lead to the same spin-entropy distribution. The probability is then as follows:
(32)$$\begin{array}{cc} P_{\theta_{1}}^{s=1}\left(u,\phi \right) & =\frac{3}{16\pi }\left(\cos^{2}\theta_{1}\;{\left(1+u\right)}^{2}\;+\sin^{2}\theta_{1}{\left(1-u\right)}^{2}+\sin2\theta_{1}\;\left(1-u^{2}\right)\cos\left(2\phi \right)\right)\\ \; & =\frac{3}{16\pi }\;\left(\left(1+u^{2}\right)+2u\;\cos\left(2\theta_{1}\right)+\sin2\theta_{1}\;\left(1-u^{2}\right)\cos\left(2\phi \right)\right) \end{array}$$
where u=cosθ. The spin-entropy of this state is as follows:
(33)$$\begin{array}{cc} S_{\theta_{1}} & =-\int_{-1}^{1}\int_{0}^{2\pi }P_{\theta_{1}}^{s=1}\left(u,\phi \right)\ln P_{\theta_{0}}^{s=1}\left(u,\phi \right)\;du\;d\phi \;. \end{array}$$The extremes of the entropy occur when $\frac{\partial S_{\theta_{1}}}{\partial \theta_{1}}=0$, i.e., when
(34)$$\begin{array}{cc} 0 & =\frac{\partial S_{\theta_{1}}}{\partial \theta_{1}}=-\int_{-1}^{1}\int_{0}^{2\pi }\frac{\partial P_{\theta_{1}}^{s=1}}{\partial \theta_{1}}\ln P_{\theta_{0}}^{s=1}\left(u,\phi \right)\;du\;d\phi \;\\ \; & =-\frac{3\;}{8\pi }\;\int_{-1}^{1}\int_{0}^{2\pi }\left(-2u\sin2\theta_{1}\;+\;\cos2\theta_{1}\;\;\left(1-u^{2}\right)\cos\left(2\phi \right)\right)\\ \; & \;\times \ln\left(\left(1+u^{2}\right)+2u\;\cos\left(2\theta_{1}\right)+\sin2\theta_{1}\;\left(1-u^{2}\right)\cos\left(2\phi \right)\right)\;. \end{array}$$They occur when the integral in ϕ vanishes due to an odd integrand function in ϕ∈[0,2π], i.e., for $\theta_{1}=0,\frac{\pi }{2}$, integrand $\left(1-u^{2}\right)\cos\left(2\phi \right)\;\ln\left(\left(1+u^{2}\right)+2u\right)$ changes signs as $\phi \to \phi +\frac{\pi }{2}$. They also occur when the integral in *u* vanishes for having an odd integrand, i.e., for $\theta_{1}=\frac{\pi }{4},\frac{3\pi }{4}$, integrand $-2u\ln(\left(1+u^{2}\right)+\left(1-u^{2}\right)\cos\left(2\phi \right))$ changes signs as u→−u. By investigation, cases $\theta_{1}=0,\frac{\pi }{2}$ are the *z*-eigenstates s=1,m=±1, and yield the minimum spin-entropy. Cases $\theta_{1}=\frac{\pi }{4},\frac{3\pi }{4}$ are associated with the *x*-eigenstates and *y*-eigenstates, $s=1,m_{x},m_{y}=0$, respectively, i.e., they are canonical transformations of the *z*-eigenstates, s=1,m=0, and yield the maximum spin-entropy. □

We show in Figure 2 a simulation for this spin-entropy as $\theta_{1}$ varies. Note that the state $\psi_{s=1,m=0}\left(\theta ,\phi \right)$ has a higher spin-entropy than the two states $\psi_{s=1,m=\pm 1}\left(\theta ,\phi \right)$, i.e., there is more randomness and less information at the m=0 state.

### 3.3. Any Spin Value

**Conjecture** **1**(Min and Max)**.**
*The spin-entropy associated with any spin-state of magnitude s has (i) its maximum value for the states $m=\pm min_{m}=\pm \mathrm{mod}\left(s,1\right)$ and all its canonical transformations, and (ii) its minimum value for the pair of states m=±s and all its canonical transformations.*

We have proven this to be true for the cases of s=1 and $s=\frac{1}{2}$ and s=0. Also, by Lemma 3, the *z*-direction eigenstates, for any *s* value, will have lower spin-entropy as $m^{+}$ increases. All canonical transformations of a state will have the same spin-entropy. It remains to be proven that any spin-state falls within the spectrum of these values, or whether the extremes of spin-entropy correspond to spin-eigenstates.

## 4. Mixed States: Von Neumann Entropy vs. Spin-Entropy

Mixed states extend the Hilbert space of specified quantum states, or pure states, to quantum states that are not fully specified, and instead are described by a classical probabilistic combination of pure states. Let a spin mixed state be defined via the density matrix, as follows:(35)$$\begin{array}{c} \rho_{s}^{M}=\sum_{i=1}^{P}\gamma_{i}|\xi_{s}^{i}\rangle \langle \xi_{s}^{i}|\;, \end{array}$$
where $\gamma_{i}\geq 0$, $1=\sum_{i=1}^{P}\gamma_{i}$, and $|\xi_{s}^{i}\rangle =\sum_{m=-s}^{s}\alpha_{s,m}^{i}|\xi_{s,m}\rangle ,\;i=1,\ldots ,P$ are pure states decomposed in spin eigenstates of the operator $S_{Z}$, satisfying $1=\sum_{m=-s}^{s}{\left|\alpha_{s,m}^{i}\right|}^{2}$. In order to quantify the randomness associated with this density matrix, we first project it to the spin phase space as follows
(36)$$\begin{array}{cc} \sum_{i=1}^{P}\rho_{s}^{M}\left(i,\theta ,\phi \right) & =\langle z\left(\theta ,\phi \right),z^{*}\left(\theta ,\phi \right)|\rho_{s}^{M}|z\left(\theta ,\phi \right),z^{*}\left(\theta ,\phi \right)\rangle \\ \; & =\sum_{i=1}^{P}\gamma_{i}\langle z\left(\theta ,\phi \right),z^{*}\left(\theta ,\phi \right)|\xi_{s}^{i}\rangle \langle \xi_{s}^{i}|z\left(\theta ,\phi \right),z^{*}\left(\theta ,\phi \right)\rangle \\ \; & =\sum_{i=1}^{P}\gamma_{i}\sum_{m=-s}^{s}\alpha_{s,m}^{i}\psi_{s,m}\left(\theta ,\phi \right)\sum_{m^{\prime }=-s}^{s}{\left(\alpha_{s,m^{\prime }}^{i}\right)}^{*}\;\psi_{s,m^{\prime }}^{*}\left(\theta ,\phi \right)\\ \; & =\sum_{i=1}^{P}\gamma_{i}\;{\left|\sum_{m=-s}^{s}\alpha_{s,m}^{i}\psi_{s,m}\left(\theta ,\phi \right)\right|}^{2}\;. \end{array}$$The quantity $\rho_{s}^{M}\left(i,\theta ,\phi \right)$, with the normalization $1=\sum_{i=1}^{P}\int_{0}^{\pi }\;\int_{0}^{2\pi }\;\rho_{s}^{M}\left(i,\theta ,\phi \right)\;d\theta \;d\phi$, is the probability of the mixed state being in a pure state $|\xi_{s}^{i}\rangle$ with observables (θ,ϕ). Extending the spin-entropy (Equation 1) to include mixed states, we define the following:(37)$$\begin{array}{cc} S^{M}\left(\rho_{s}^{M}\left(i,\theta ,\phi \right)\right) & =-\sum_{i=1}^{P}\int_{0}^{\pi }\;\int_{0}^{2\pi }\;\rho_{s}^{M}\left(i,\theta ,\phi \right)\ln\rho_{s}^{M}\left(i,\theta ,\phi \right)\;d\theta \;d\phi \\ \; & =S^{\mathrm{vN}}+\sum_{i=1}^{P}\gamma_{i}S_{s}^{i}\left(\rho_{s}\left(\theta ,\phi |i\right)\right)\;, \end{array}$$
where von Neumann entropy is $S^{\mathrm{vN}}=-\sum_{i=1}^{P}\gamma_{i}\ln\gamma_{i}$, and the conditional entropy of each pure state in the mixture is $S_{s}^{i}\left(\theta ,\phi |i\right)=-\int_{0}^{\pi }\;\int_{0}^{2\pi }\;\rho_{s}\left(\theta ,\phi |i\right)\ln\rho_{s}\left(\theta ,\phi |i\right)\;\;d\theta \;d\phi$, with the conditional probability given by $\rho_{s}\left(\theta ,\phi |i\right)=\frac{\rho_{s}^{M}\left(i,\theta ,\phi \right)}{\gamma_{i}}={\left|\sum_{m=-s}^{s}\alpha_{s,m}^{i}\psi_{s,m}\left(\theta ,\phi \right)\right|}^{2}$. The spin-entropy is always larger than von Neumann entropy since we also add the expected value of the spin-entropy of the *P* pure states, which are all positive for all known physical particles (see Lemma (4)).

## 5. Phase Space Entanglement Increases Entropy

Consider two orthonormal spin eigenstates along the *z*-direction, $\psi_{s,m_{1}}\left(\theta \right)\;e^{i\;(s-m_{1})\phi }$, and $\psi_{s,m_{2}}\left(\theta \right)\;e^{i\;(s-m_{2})\phi }$, where from (Equation 12), we have the following:(38)$$\begin{array}{cc} \psi_{s,m_{1,2}}\left(\theta \right) & =\frac{{\left(2s+1\right)}^{\frac{1}{2}}}{\sqrt{4\pi }}\;{\left[C_{s+m_{1,2}}^{2s}\right]}^{\frac{1}{2}}{\left(\cos\frac{\theta }{2}\right)}^{s+m_{1,2}}{\left(\sin\frac{\theta }{2}\right)}^{s-m_{1,2}} \end{array}$$Following [7,8,24,25,26] and their references, let us define the entanglement of these two orthonormal spin eigenstates as follows:(39)$$\begin{array}{cc} \Psi_{s,m_{1},m_{2}}\left(\theta ,\phi ,\theta^{\prime },\phi^{\prime };\theta^{e}\right) & =\cos\theta^{e}\psi_{s,m_{1}}\left(\theta \right)\psi_{s,m_{2}}\left(\theta^{\prime }\right)\;e^{i[\left(s-m_{1}\right)\phi +\left(s-m_{2}\right)\phi^{\prime }]}\\ \; & \;+\sin\theta^{e}\psi_{s,m_{2}}\left(\theta \right)\psi_{s,m_{1}}\left(\theta^{\prime }\right)\;e^{i[\left(s-m_{2}\right)\phi +\left(s-m_{1}\right)\phi^{\prime }]}\;, \end{array}$$
where $\theta^{e}\in \left[0,\pi \right)$ periodically controls the amount of entanglement, with minimum (no) entanglement occurring at $\theta^{e}=0,\frac{\pi }{2}$ and maximum entanglement occurring at $\theta^{e}=\frac{\pi }{4},\frac{3\pi }{4}$.

By Born’s rule, this spin-state has a probability density, as follows:(40)$$\begin{array}{cc} P_{s,m_{1},m_{2}}\left(\theta ,\phi ,\theta^{\prime },\phi^{\prime };\theta^{e}\right) & =|\Psi_{s,m_{1},m_{2}}\left(\theta ,\phi ,\theta^{\prime },\phi^{\prime };\theta^{e}\right){|}^{2}\;, \end{array}$$The spin-entropy of state (Equation 39) is as follows: (41)$$\begin{array}{cc} S_{s,m_{1},m_{2}}\left(\theta^{e}\right) & =-\int_{-1}^{1}\int_{0}^{2\pi }\int_{-1}^{1}\int_{0}^{2\pi }P_{s,m_{1},m_{2}}\left(u,\phi ,u^{\prime },\phi^{\prime };\theta^{e}\right)\ln P_{s,m_{1},m_{2}}\left(u,\phi ,u^{\prime },\phi^{\prime };\theta^{e}\right)du\;d\phi \;du^{\prime }\;d\phi^{\prime }\;. \end{array}$$The extremes of the entropy occur when $\frac{\partial S_{s,m_{1},m_{2}}\left(\theta^{e}\right)}{\partial \theta^{e}}=0$, i.e., when
(42)$$\begin{array}{cc} 0 & =\frac{\partial S_{s,m_{1},m_{2}}\left(\theta^{e}\right)}{\partial \theta^{e}}\\ \; & =-\int_{-1}^{1}\int_{0}^{2\pi }\int_{-1}^{1}\int_{0}^{2\pi }\frac{\partial P_{s,m_{1},m_{2}}\left(u,\phi ,u^{\prime },\phi^{\prime };\theta^{e}\right)}{\partial \theta^{e}}\ln P_{s,m_{1},m_{2}}\left(u,\phi ,u^{\prime },\phi^{\prime };\theta^{e}\right)\;du\;d\phi \;du^{\prime }\;d\phi^{\prime }\;. \end{array}$$We conjecture that they occur for $\theta^{e}=0,\frac{\pi }{4},\frac{\pi }{2},\frac{3\pi }{4}$. By investigating these four cases, the cases $\theta^{e}=0,\frac{\pi }{2}$, associated with no entanglement, yield the minimum spin-entropy, which is the sum of the spin-entropy of each state. We already calculated each state spin-entropy in (Equation 17). The cases $\theta^{e}=\frac{\pi }{4},\frac{3\pi }{4}$ yield the maximum spin-entropy and are associated with maximum entanglement.

We present “a sketch of a proof” by reasoning about randomness. Let us examine the role of each of the wave functions, $\psi_{s,m_{1}}\left(\theta \right)\;e^{i\;(s-m_{1})\phi }$, and $\psi_{s,m_{2}}\left(\theta^{\prime }\right)\;e^{i\;\left(s-m_{2}\right)\phi^{\prime }}$, involved in the entanglement (Equation 39). Each represents a distribution in a two-sphere $S^{2}$, or let us call it a “container”. The phase space of the entanglement is the product of these two containers. For $\theta^{e}=0,\frac{\pi }{2}$, when there is no entanglement, despite the phase space being the product of two containers, each distribution occupies only one of the containers. Thus, the entanglement spin-entropy is the sum of the spin-entropy of each container. For $\theta^{e}=\frac{\pi }{4},\frac{3\pi }{4}$, when there is maximum entanglement, the wave functions equally occupy both containers, i.e., they are mixed in the containers. As a distribution spreads wider, its entropy increases. The parameter $\theta^{e}$ controls the mixture and, hence, the more mixed the distribution, the larger the spin-entropy. This reasoning is not specific about spin, it also applies to position entanglement.

In Figure 3, we plot the spin-entropy (Equation 41) vs. $\theta^{e}$ for *z* eigenstates of $s=\frac{1}{2}$ and for *z* eigenstates of s=1. Clearly, spin-entropy increases as the entanglement increases. More analysis is offered in the figure’s caption.

## 6. Spin Interaction and Oscillations of the Spin-Entropy

We investigate the evolution of the spin-entropy for a massive particle with spin s=1 in an initial state $\psi_{s=1,m=1,\omega_{0}}^{\hat{n}}\left(\theta ,\phi ,t_{0}\right)$, where $\hat{n}$, is a unit vector indicating an arbitrary 3D direction. The state $\psi_{s=1,m=1,\omega_{0}}^{\hat{n}}\left(\theta ,\phi ,t_{0}\right)$ is not only a spin eigenstate along the direction $\hat{n}$ with m=1, but also an eigenstate of the Hamiltonian $H^{0}$ with the energy eigenvalue, $\hbar \omega_{0}=m_{0}c^{2}$, where $m_{0}$ is the rest mass.

Let us say that a magnetic field $B=B\hat{z}$ interacts with this spin-state via an interaction Hamiltonian $H^{I}=-\gamma BS_{z}$ with eigenvalues $\hbar \;m\;\omega_{\gamma }$, for the *Z*-axis eigenstates $\psi_{s=1,m=0,\pm 1}\left(\theta ,\phi \right)$, respectively, and where $\omega_{\gamma }=\frac{\gamma B}{\hbar }$

The Hamiltonians based on $\left\{|\psi_{s=1,m_{k}}^{\hat{n}}\rangle ;\;k=1,2,3\right\}$, with $m_{k}=1,0,-1$ for k=1,2,3, are as follows:(43)$$\begin{array}{cc} H^{0} & =\left(\begin{array}{ccc} \hbar \omega_{0} & 0 & 0\\ 0 & \hbar \omega_{0} & 0\\ 0 & 0 & \hbar \omega_{0} \end{array}\right)\\ {\left(H^{I}\right)}_{i,j} & =\langle \psi_{s=1,m_{i}}^{\hat{n}}|H^{I}|\psi_{s=1,m_{j}}^{\hat{n}}\rangle =\sum_{k,k^{\prime }}\langle \psi_{s=1,m_{i}}^{\hat{n}}|\psi_{s=1,m_{k}}\rangle \langle \psi_{s=1,m_{k}}|H^{I}|\psi_{s=1,m_{k}^{\prime }}\rangle \langle \psi_{s=1,m_{k}^{\prime }}|\psi_{s=1,m_{j}}^{\hat{n}}\rangle \\ \; & ={\left[U\left(\hat{n}\right)\;\left(\begin{array}{ccc} -\hbar \omega_{\gamma } & 0 & 0\\ 0 & 0 & 0\\ 0 & 0 & \hbar \omega_{\gamma } \end{array}\right)\;U^{†}\left(\hat{n}\right)\right]}_{ij} \end{array}$$
where $U_{ij}\left(\hat{n}\right)=\langle \psi_{s=1,m_{i}}^{\hat{n}}|\psi_{s=1,m_{j}}\rangle$, $m_{k}=1,0,-1$ for k=1,2,3, and $I_{3\times 3}=U\left(\hat{n}\right)\;U^{†}\left(\hat{n}\right)$.

Writing the initial state evolution on the basis of $\left\{|\psi_{s=1,m_{k}}^{\hat{n}}\rangle ;\;k=1,2,3\right\}$,
(44)$$\begin{array}{cc} \left(\begin{array}{c} \alpha_{1}\left(t\right)\\ \alpha_{0}\left(t\right)\\ \alpha_{-1}\left(t\right) \end{array}\right) & =\exp\left[\frac{-iH_{\mathrm{Tot}}t}{\hbar }\right]\;\left(\begin{array}{c} 1\\ 0\\ 0 \end{array}\right)\\ \; & =U\left(\hat{n}\right)\;\left(\begin{array}{ccc} e^{-i(\omega_{0}+\omega_{\gamma })t} & 0 & 0\\ 0 & e^{-i\omega_{0}t} & 0\\ 0 & 0 & e^{-i(\omega_{0}-\omega_{\gamma })t} \end{array}\right)\;U^{†}\left(\hat{n}\right)\;\left(\begin{array}{c} 1\\ 0\\ 0 \end{array}\right)\\ \; & =e^{-i\omega_{0}t}\;U\left(\hat{n}\right)\;\left(\begin{array}{c} U_{11}^{*}\left(\hat{n}\right)\;e^{-i\omega_{\gamma }t}\\ U_{12}^{*}\left(\hat{n}\right)\\ U_{13}^{*}\left(\hat{n}\right)\;e^{i\omega_{\gamma }t} \end{array}\right)\\ \; & =e^{-i\omega_{0}t}\left(\begin{array}{c} 1+{\left|U_{11}\left(\hat{n}\right)\right|}^{2}\;\left(e^{-i\omega_{\gamma }t}-1\right)+{\left|U_{13}\left(\hat{n}\right)\right|}^{2}\;\left(e^{i\omega_{\gamma }t}-1\right)\\ U_{21}\left(\hat{n}\right)U_{11}^{*}\left(\hat{n}\right)\;\left(e^{-i\omega_{\gamma }t}-1\right)+U_{23}\left(\hat{n}\right)U_{13}^{*}\left(\hat{n}\right)\;\left(e^{i\omega_{\gamma }t}-1\right)\\ U_{31}\left(\hat{n}\right)U_{11}^{*}\left(\hat{n}\right)\;\left(e^{-i\omega_{\gamma }t}-1\right)+U_{33}\left(\hat{n}\right)U_{13}^{*}\left(\hat{n}\right)\;\left(e^{i\omega_{\gamma }t}-1\right) \end{array}\right)\\ \; & =e^{-i\omega_{0}t}\left(\begin{array}{c} 1-\left({\left|U_{11}\left(\hat{n}\right)\right|}^{2}+{\left|U_{13}\left(\hat{n}\right)\right|}^{2}\right)\;\sin\frac{\omega_{\gamma }t}{2}-i\left({\left|U_{11}\left(\hat{n}\right)\right|}^{2}-{\left|U_{13}\left(\hat{n}\right)\right|}^{2}\right)\;\sin\omega_{\gamma }t\\ -\left(U_{21}\left(\hat{n}\right)U_{11}^{*}\left(\hat{n}\right)+U_{23}\left(\hat{n}\right)U_{13}^{*}\left(\hat{n}\right)\right)\;\sin\frac{\omega_{\gamma }t}{2}-i\left(U_{21}\left(\hat{n}\right)U_{11}^{*}\left(\hat{n}\right)-U_{23}\left(\hat{n}\right)U_{13}^{*}\left(\hat{n}\right)\right)\;\sin\omega_{\gamma }t\\ -\left(U_{31}\left(\hat{n}\right)U_{11}^{*}\left(\hat{n}\right)+U_{33}\left(\hat{n}\right)U_{13}^{*}\left(\hat{n}\right)\right)\;\sin\frac{\omega_{\gamma }t}{2}-i\left(U_{31}\left(\hat{n}\right)U_{11}^{*}\left(\hat{n}\right)-U_{33}\left(\hat{n}\right)U_{13}^{*}\left(\hat{n}\right)\right)\;\sin\omega_{\gamma }t \end{array}\right)\;. \end{array}$$The state of the particle at any time, *t*, described by a superposition of the three states, $\left\{|\psi_{s=1,m_{k}}^{\hat{n}}\rangle ;\;k=1,2,3\right\}$, with superposition coefficients $\alpha_{m_{k}}\left(t\right)$, will oscillate over time with a period of $T=\frac{\pi }{\omega_{\gamma }}$. Therefore, the entropy of the superposition will also oscillate with the same period.

These calculations are similar to the ones employed to calculate Fermi’s golden rule [27,28]. However, here, $H^{0}$ yields the same energy for all considered spin states.

## 7. Conclusions

This paper quantifies the randomness associated with a spin-state by defining the spin-entropy in phase space.

At the formal level, we use differential entropy from information theory and the geometric quantization method (GQ) to the two-sphere. GQ describes the spin originating from classical ideas, and it also provides the spin phase space where the spin wave function and its randomness are defined. Local canonical transformation in phase space preserves the area element of the sphere and guarantees that the spin-entropy satisfies the homogeneity hypothesis; thus, Lemma 1 shows that the spin-entropy of a spin-state is invariant under 3D rotations and reflections of the coordinate system. We extend the spin-entropy to mixed states and show that von Neumann entropy is a lower bound for it.

Lemma 2 provides a spin-entropy formula for the *z*-direction spin-eigenstates. Lemma 3 and a plot illustrating the spin-entropy of a given spin, *s*, versus different values of m=−s,−s+1,…,s−1,s, see Figure 1, show that spin-entropy exhibits symmetry with respect to m↔−m, and decreases as *m* increases for m≥0. Thus, in particular, the state $\psi_{s=1,m=0}\left(\theta ,\phi \right)$ has a higher spin-entropy than the spin states $\psi_{s=1,m=\pm 1}\left(\theta ,\phi \right)$, i.e., there is more randomness and less information at the m=0 state.

The phase space of the entanglement of two states is the product of two phase spaces. We reasoned that entanglement corresponds to a distribution that mixes the two phase spaces while the product of the states does not mix. The larger the entanglement, the more the mixture, the more randomness, and the larger the entropy. We speculate that the phenomena of decoherence [7,8,24,26,29] of a subsystem immersed in an environment occur, so that parts of such a subsystem will entangle with the environment, and the total entanglement increases. One interesting future direction could be to exploit a connection between this work and the works on quantum thermalization [10,30,31,32], as well as their references; these references suggest a procedure that—by tracing out the environment and evaluating the reduced density matrix of a system of interest—may lead to the entropy of classical physics.

We investigate a dynamic model of an interaction Hamiltonian of a constant magnetic field with the spin, given an initial superposition of states. We show that the time evolution of the spin-entropy oscillates. Investigations into the hypothesis proposed by Geiger and Kedem [15] show that the quantum entropy of the closed physical system cannot decrease, implying that information from a closed system cannot be gained, and will be left for future work.

## Figures and Tables

**Figure 1 entropy-26-00372-f001:**
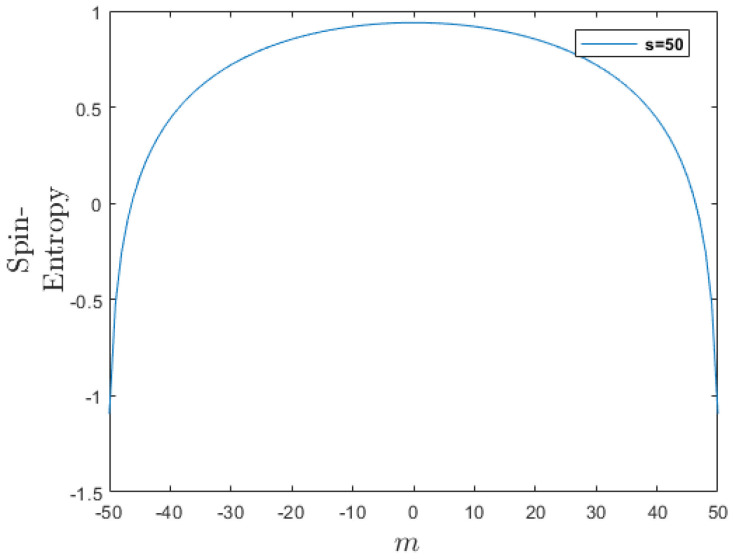
Spin-entropy for s=50 eigenstates along the *z*-direction. It is invariant by transforming m→−m and it decreases as *m* increases for m≥0 (see Lemma 3).

**Figure 2 entropy-26-00372-f002:**
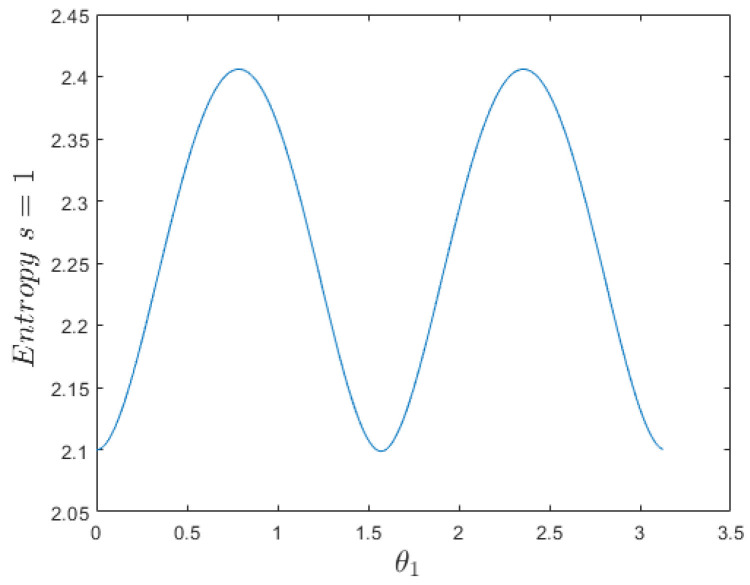
Spin-entropy (Equation 33) vs. $\theta_{1}\in \left[0,\pi \right)$ for the s=1 state (Equation 31), a superposition of $\psi_{s=1,m=1}\left(\theta ,\phi \right)$, and $\psi_{s=1,m=-1}\left(\theta ,\phi \right)$. The extremes occur for $\theta_{1}=0,\frac{\pi }{4},\frac{\pi }{2},\frac{3\pi }{4}$. At its maximum $\theta_{1}=\frac{\pi }{4},\frac{3\pi }{4}$, the superposition of states (Equation 31) is a canonical transformation of a state s=1,m=0, either to the *x*-eigenstate $s=1,m_{x}=0$ ($\theta_{1}=\frac{\pi }{4}$) or to the *y*-eigenstate $s=1,m_{y}=0$ ($\theta_{1}=\frac{3\pi }{4}$).

**Figure 3 entropy-26-00372-f003:**
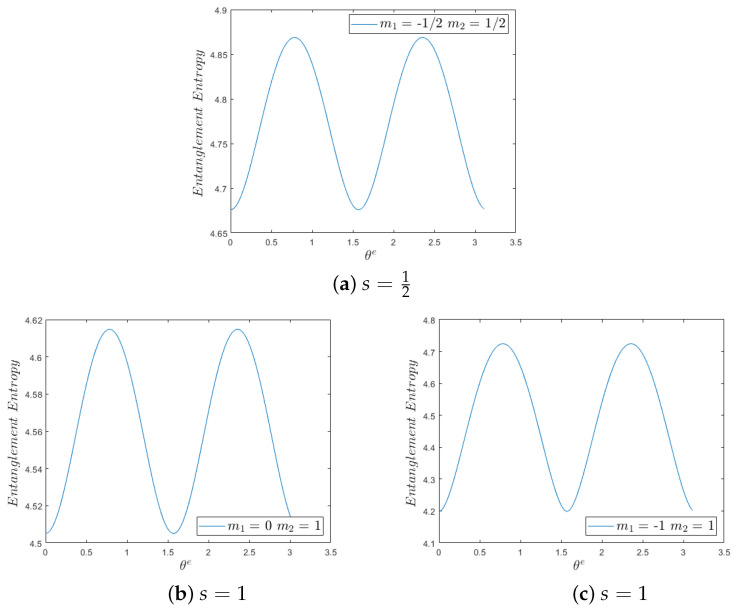
Plots of spin-entropy of entanglement (Equation 41) vs. $\theta^{e}$, a parameter that controls the amount of the entanglement. When $\theta^{e}=0,\frac{\pi }{2}$, there is no entanglement (product state), and when $\theta^{e}=\frac{\pi }{4},\frac{3\pi }{4}$, there is maximum entanglement. In all graphs, the spin-entropy increases as the amount of the entanglement increases. (**a**). The spin-entropy of the entanglement $s=\frac{1}{2}$, for any given $\theta^{e}$, is larger than the spin-entropy of the entanglements shown in b. and c., for the same $\theta^{e}$. (**b**). The entanglement entropy shown for $m_{1}=0,m_{2}=1$ is the same as in the case $m_{1}=0,m_{2}=-1$. This can be inferred from the mapping, $m_{1}=0,m_{2}=1\to m_{1}=0,m_{2}=-1$ being described by the mapping $\left(\theta ,\theta^{\prime }\right)\to \left(\pi -\theta ,\pi -\theta^{\prime }\right)$ which leads to the same probabilities (Equation 40) and, thus, to the same spin-entropy (Equation 41). (**c**). The minimum entanglement value is less than in b., after all in these cases the spin-entropy is simply the sum of the spin-entropy of both states. The maximum entanglement value is more than in b. We wonder if it may be related to the fact that the superposition of these two states is a canonical transformation of the higher entropy state $\psi_{s=1,m=0}$; see Figure 2.

## Data Availability

Data are contained within the article.

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
