# Peer review of "Phase Space Spin-Entropy"

_entropy, 2024, doi:10.3390/e26050372_

Round 1

Reviewer 1 Report

Comments and Suggestions for Authors Report on Ms. entropy-2954794 The author addresses an important question of general relevance for understanding quantum spin systems. The mathematical apparatus is well formulated and the results on the spin entropy will be of interest for mathematical physics. However, the attempt to use the weak gauge bosons as example for a spin-1 entropic system is in danger of creating misunderstandings if not indicating a misconception. Each of the W+, W-, and the Z bosons are by themselves (ordinary) spin-1 particles. They are NOT being considered as the m=+1,-1,0 members of a super-spin triplet object. In the light meson sector, the rho-meson as a spin-1, isospin-1 particle comes closest to the case discussed by the author, as far as a (partially) conserved intrinsic symmetry is concerned. The author must clarify this point before possibly publication can be recommended.

Author Response

Thank you. 

" ...However, the attempt to use the weak gauge bosons as example for a spin-1 entropic system is in danger of creating misunderstandings if not indicating a misconception."

I understand you, thank you. I believe my view was not well expressed nor quantified. So I rewrote this material creating a section for it, attached section 5.  Below, I am following on it as a dialogue asking your feedback

" Each of the W+, W-, and the Z bosons are by themselves (ordinary) spin-1 particles. They are NOT being considered as the m=+1,-1,0 members of a super-spin triplet object."

Yes. However, because they are elementary particles, I think my hypothesis can be better formalized. I mean,  m=+1,-1,0, is simply referring to the z component of the spin, and that can changed if a magnetic field is "surrounding/interacting" with such particles (like it does on the early  Stern-Gerlach experiments). Yes?

"...In the light meson sector, the rho-meson as a spin-1, isospin-1 particle comes closest to the case discussed by the author, as far as a (partially) conserved intrinsic symmetry is concerned. .."

I see your point and that would be interesting to explore in the future. In this paper, I am trying to "stay" as close as possible to the elementary particles behavior, because I do believe that such spin-entropy does play a role on elementary particles (and consequently on all composed particles). Note that these gauge bosons are the only massive gauge bosons, i.e., the only ones  with all possible spin z components.  Sure enough, there is no decay for them. Of course, it is today just a hypothesis.  

" ...The author must clarify this point before possibly publication can be recommended."

Attached is a a new version of the article with section 5 added to better address this topic and quantify the period of decay. 

Reviewer 2 Report

Comments and Suggestions for Authors

The paper is nice.  

One could argue (in the introduction) that quantum physics is intrinsically probabilistic. Quantum evolution is  deterministic, the Born's rule is  probabilistic.

Author Response

Thank you.  What about 

" Quantum physics is intrinsically probabilistic, where Born's rule yield the probabilities associated with a state that deterministically evolves over time. ..."?